# How Buddhist Religious Care Is Incorporated for End-of-Life Stroke Patients Receiving Palliative Care at Home during the COVID-19 Pandemic: Revisiting Constructivist Grounded Theory

**Jinpitcha Mamom** [1,2] and **Hanvedes Daovisan** [3,*]

1. Center of Excellence in Creative Engineering Design and Development, Faculty of Engineering, Thammasat University, Pathumthani 12120, Thailand
2. Department of Adult Nursing and the Aged, Faculty of Nursing, Thammasat University, Pathumthani 12120, Thailand
3. Human Security and Equity Research Unit, Social Research Institute, Chulalongkorn University, Bangkok 10330, Thailand
* Correspondence: hanvedes.d@chula.ac.th

**Abstract:** Coping with the COVID-19 pandemic has involved unprecedented health challenges, impacting not only the receipt of palliative care, but also that of religious care. The present article aimed to explore how Buddhist religious care is incorporated for end-of-life stroke patients receiving palliative care at home during the COVID-19 pandemic in Thailand. A constructivist grounded theory (CGT) was utilised with theoretical sampling of 30 respondents in the Angthong, Ayutthaya, and Pratumthani provinces, central Thailand, from June 2020 to March 2022. Online in-depth interviews were video-recorded and transcribed, using written memos and constant comparative methods. Data transcripts were analysed using open, axial, selective coding, categorising, and themes. Our CGT study identified five themes of Buddhist religious care incorporated for end-of-life stroke patients receiving palliative care at home, namely Buddhist therapies, religious beliefs, religious life satisfaction, religious mental care, and religious needs. The implications of Buddhist religious care being incorporated for end-of-life stroke patients receiving home palliative care during the COVID-19 pandemic are discussed.

**Keywords:** Buddhist religious care; palliative care; end-of-life; stroke; family caregivers; COVID-19; Thailand

## 1. Introduction

Thailand's first wave of the coronavirus disease 2019 (COVID-19) began on 26 March 2020, rapidly spreading across the country, following which the government called for a strong combat mechanism with which to tackle the pandemic (Thambhitaks and Boonyathee 2021). COVID-19 has led to substantial disruption of the healthcare system and delivery of health services, whilst it has also caused medical shortages. The disease has wreaked havoc, giving rise to a growing death toll, severe acute respiratory syndrome (SARS), and affecting the mental health of clinicians. With the spread of COVID-19 to rural areas, and the overloading of health facilities, there has been a need to protect vulnerable palliative care patients from contracting the virus. As such, patients, caregivers, healthcare providers, and health systems are in severe need of religious ways in which to cope with the COVID-19 pandemic (Sukcharoen et al. 2020). Additionally, family caregivers have tended to view religious elements as a cure for their patients' illness (Chaiviboontham and Pokpalagon 2021; Connolly and Timmins 2022; Mamom and Daovisan 2022) and have played an important role in incorporating religious intervention into palliative care settings (Cetty et al. 2022; Gijsberts 2022).

This is in line with literature showing that palliative care is the patient's favourite setting when it comes to religiously coping with the COVID-19 pandemic (Counted et al. 2022; Wilt et al. 2022). The impact of COVID-19 has led to a surge in demand for palliative care (Marshall et al. 2021). In such palliative care settings, end-of-life stroke patients need to be treated at home, where they can be cared for by their families and relatives. The term palliative care at home is defined as "the primary relief from pain, symptoms, and support quality of life for patients with serious illness" (Krongyuth et al. 2014). Nilmanat (2016) classified palliative care at home as follows: (i) complete care of mind, soul, and body, (ii) support family caregivers, (iii) tertiary prevention plan, and (iv) coordinator of care in the unit. According to Chaiviboontham and Pokpalagon (2021), palliative care at home is related to the delivery of healthcare and mental counselling services for patients.

Thailand has endorsed the rights of stroke patients receiving palliative care at home (Phungrassami et al. 2013), but practices are still unclear. When faced with COVID-19, however, little is known about how religious care is incorporated into palliative care at home. Past studies indicated that palliative care at home is dealt with via religious coping, decision-making, psychological support, consultation for relatives, and ritual care. Jung et al. (2022) and Voytenko et al. (2021) suggested that, in non-Western countries, the religious care is the extent to which people have subjective beliefs, follows, and practices. Positive religious coping, which includes strategies for seeking Buddhist religious support (Smith-Stoner 2006), is associated with less stroke-related pain (Dorji and Lapierre 2022) in end-of-life patients receiving palliative care at home.

Central Thailand is one of the country's most religious regions, with more than 95% of patients being Buddhists (Ministry of Public Health 2004). Incorporating Buddhist religious care for end-of-life patients is related to follows, teaching, and practices (Kongsuwan et al. 2010; Sethabouppha and Kane 2005; Strong 2021). Although there are Buddhists in almost all palliative care settings, religious care may differ in various parts of the family who are acting as caregivers. Past studies identified three kinds of Buddhist care: care for the sick, medical care and caregiving, and care for the dying. As mentioned previously, according to Smith-Stoner (2006), incorporating Buddhist religious care is associated with belief system (basic practices), personal religiosity (learn how the patient functions), integration into a religious community (activities in the religiosity), ritualised practice and restrictions (a basic creed of Buddhism), implications for medical care (advance directives), and terminal event planning (read the scripts to the patient).

Previous studies have examined the Buddhist-orientated religious coping: meaning-making coping, meditative coping, and ego-transcendence coping (Xu 2021). Some scholars further assessed meaning in life (Zhang et al. 2021), intensity of prayer in Buddhist practices (Bentzen 2021), and attainment of merit and mental well-being (Benoit et al. 2021). In response to religious coping with COVID-19, family caregivers have been implicated in fostering Buddhist mantra repetition to enhance quality of life, self-efficacy, and mindfulness (Oman et al. 2022). In terms of the COVID-19 pandemic, Buddhist practices are associated with meditation, care partners, and reprioritising one's life (Soonthornchaiya 2020). For Buddhist patients, family caregivers are a meditative influence and the most important support mechanism when it comes to coping with the COVID-19 pandemic.

Now focusing on the palliative care perspective, Buddhist religious care has a long history in terms of family caregiver practices: there is repetition of a single short word, phrase, or prayer—commonly known as the Buddhist practice (Chaiviboontham and Pokpalagon 2021). Amongst Thai Buddhist family caregivers, Upasen et al. (2022) stated that repeated praying and sermons, listening to Dharma, meditation, practice, and mindfulness were the key attributes of the end-of-life care practices. Kalra et al. (2018) highlighted that Buddhist family caregivers are defined according to three levels: the Buddha (the religious guide), the Dharma (the practice), and the Sangha (the community). This study addressed restrictions for palliative care at home, especially in Thailand, where practice and communication about patients and caregivers at the end-of-life stage has increasingly become the norm.

This is important because Buddhist religious care has been incorporated for end-of-life stroke patients receiving palliative care at home during the COVID-19 pandemic. There is growing evidence regarding the effectiveness of Buddhist religious care due to healthcare system disruption (Counted et al. 2022; Diego-Cordero et al. 2022; Domaradzki 2022; Dorji and Lapierre 2022; Gilissen et al. 2020; Kwak et al. 2022; Marshall et al. 2021; Oman et al. 2022; Soonthornchaiya 2020; Thambhitaks and Boonyathee 2021; Voytenko et al. 2021), but Buddhist religious care was not commonly incorporated at the onset of the COVID-19 pandemic. Whilst various studies have focused on Buddhist practices (Stewart et al. 2016), we explored how Buddhist religious care has been incorporated for end-of-life stroke patients receiving palliative care at home in central Thailand.

*Research Questions*

The present article aimed to explore how Buddhist religious care has been incorporated for end-of-life stroke patients receiving palliative care at home during the COVID-19 pandemic in central Thailand. To achieve this, we focused on: (i) important aspects of religious care, (ii) religious beliefs, (iii) personal religious life, (iv) religious mentality, and (v) religious medical care. The key research questions are as follows:

*Research Question 1*: What are the most important aspects of religious care involving end-of-life stroke patients at home during the COVID-19 pandemic?
*Research Question 2*: How is Buddhist religious care incorporated into palliative care at home during the COVID-19 pandemic?

## 2. Methods

### 2.1. CGT Design

This article utilised the CGT design (Charmaz 2006), as can be seen in Figure 1. The design is theoretically generated with a grounded approach to insider and outsider respondent experiences, alongside the existing theory informing the field. We used this technique because it allows us to discover grounded data, seeks to understand the Buddhist religious care, and constructs theories of palliative care. This approach made it possible for the researchers to explore the impact of COVID-19 disruption on the healthcare system from the perspective of end-of-life stroke patients receiving palliative care at home in central Thailand. We preferred the CGT approach to underpin (i) the gathering of data, (ii) coding, (iii) analytical memo writing, (iv) theoretical sampling and sorting, and (v) emerging theory.

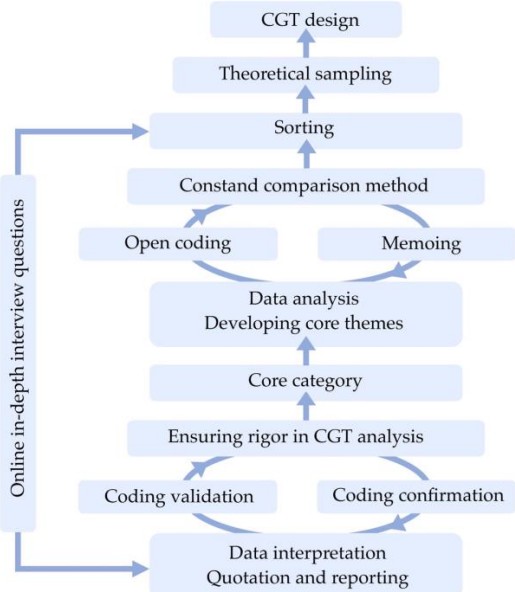

**Figure 1.** CGT design.

## 2.2. CGT Setting

This article explored family caregivers for end-of-life stroke patients receiving palliative care at home in the Angthong, Ayutthaya, and Pratumthani provinces, central Thailand. The locations experienced five waves of COVID-19 transmission in March 2020, December 2020, February 2021, June 2021, and December 2021. We selected these three sites because they are the largest and most dense in terms of the number of end-of-life stroke patients receiving palliative care at home in the communities of central Thailand. The respondents were eligible if they were: (i) bedridden stroke patients, (ii) family caregivers providing day-to-day care, (iii) were not financially reimbursed for caregiving activities, (iv) had been cared for over a period of 15 months prior to the study, (v) had a medical diagnosis of moderate and severe stroke, (vi) were aged 60 years and over, and (vii) believed in Buddhist care and caregiving (Smith-Stoner 2006). Respondents (whether patients or family caregivers) dealing with dementia, schizophrenia, manic-depressive disorder, and intellectual disabilities were excluded.

## 2.3. Ethical Considerations

The Human Research Ethics Committee of Thammasat University (Science), with which the first author is affiliated, granted ethical approval for this study. The current article was approved by the Thammasat University and the Declaration of Helsinki, the Belmont Report, CIOMS Guidelines and International Practice (ICH-GCP) (COA No. 117/2562, Ref No. 065/2562). All respondents were informed of the research objective, participated voluntarily, and signed an informed consent document. The respondents' anonymity, privacy, confidentiality, and personalised information were guaranteed at all stages of the study. All data were de-identified and reported only at an aggregate level.

## 2.4. CGT Theoretical Sampling and Participants

Utilising theoretical sampling (Charmaz 2006), respondents were asked to identify suitable family caregivers of end-of-life stroke patients at home in central Thailand. The dataset of patients aged over 60 years, who were surveyed during the period spanning June 2020 to March 2022, consisted of: 890 patients in Angthong Province, 1050 in Ayutthaya Province, and 1110 in Pratumthani Province (Long Term Care 2020). Initially, purposive sampling was used to recruit 100 bedridden stroke patients from each of the three above-mentioned provinces. Secondly, concept mapping helped the researcher to select 50 further relevant patients per each province, all of whom were bedridden and receiving stroke care at home. Thirdly, theoretical sampling was employed to recruit 25 respondents per each province, with their condition ranging from moderate stroke (score = 15–20) to severe stroke (score = 21–42). Finally, respondent selection for generating theory equally selected 30 end-of-life stroke patients (10 respondents per each province) who were defined as receiving palliative care at home from family caregivers.

## 2.5. CGT Online In-Depth Interview Questions

Online in-depth interview questions asked about important aspects of religious care, religious beliefs, personal religious life, religious mentality, and religious medical care (see Table 1). Interviews ranged from 60 to 120 minutes per respondent and were conducted during the period spanning June 2020 to March 2022, with Thai being the only language used. The questions were elaborated on through probing queries such as "how does your…how do you…how would you…what do you think". The longitudinal online in-depth interviews were applied in order to explore the nature and degree of changes in family caregivers for end-of-life stroke care during the COVID-19 pandemic. We re-interviewed the same respondents in four rounds: first, from June to November 2020; second, from December 2020 to February 2021; third, from June to November 2021; and finally, from January to March 2022.

**Table 1.** Sample online in-depth interview questions.

| Issues | Sub-Interview Questions |
| --- | --- |
| Introduction | Welcome note, guidelines, and purpose of the online interviews |
| General questions | What are the characteristics and stroke levels of your end-of-life stroke patient? |
| | How does your religious coping involve end-of-life stroke care? |
| Important aspects of religious care | How does your religious advice enhance end-of-life stroke care? |
| | How does your religious advice ease suffering during end-of-life stroke care? |
| | How does your eternal life and faith encourage the patient during end-of-life stroke care? |
| Religious beliefs | How is your religious life incorporated for end-of-life stroke care? |
| | What do you think about the nature of reality for end-of-life stroke care? |
| | How would you describe mental living in religious care? |
| Personal religious life | How do you feel about moment-to-moment in religious care? |
| | How do you find meaning in religious care? |
| | How does your feeling connectedness play out in religious care? |
| | How do you incorporate sensations into a religious mentality? |
| Religious mentality | How do you enhance recognition in a religious mentality? |
| | How do you provide desires in a religious mentality? |
| | How do you improve consciousness in a religious mentality? |
| | How do you relieve pain during end-of-life stroke care? |
| Religious medical care | How do you incorporate physiological stimulus for end-of-life stroke care? |
| | How does your religious healing benefit end-of-life stroke care? |

### 2.6. CGT Analyses

All transcripts were analysed using the CGT approach (open coding, axial coding, selective coding, categories, concepts, and themes) (Mills et al. 2006). Open coding involved line-by-line coding of categories with a similar meaning, which were grouped by high-level codes and categories. Axial coding was applied to the categories (Strauss and Corbin 2008), whilst selective coding was used to derive a core category for developing themes. The developing themes created a conceptual description of conceptual development. Emerging theory was confirmed and generated the relationship with the constructed theory.

### 2.7. CGT Rigour

The following are issues related to the rigorous CGT approach: validity, reflexivity, auditability, and fitness (Chiovitti and Piran 2003). To ensure validity, we re-interviewed, employed constant comparison methods, used coding strategies, checked the verifications, and came up with theories. The interviews were recorded to confirm reflexivity in the interaction between researchers and respondents, which reduces potential bias. For auditability, all records were transparent, allowing for scrutiny and review. To ensure fitness, we conducted theoretical sampling and then re-conducted it so as to obtain the suitable respondent experiences in Buddhist religious care. The data triangulation was validated by the first author and co-authors' peer revision of the dataset. To improve the accuracy, the results were also read by research teams who are experts in the field. All processes were adopted from Bryant's (2017) rigorous grounded theory research: (i) rich data, (ii) respondent validity, (iii) triangulated data sources, and (iv) comparison (the writing up of the researcher's reflections).

### 3. Results

A total of 30 respondents were recruited from family caregivers of end-of-life stroke patients receiving palliative care at home during the COVID-19 pandemic. The average care duration was between 15 and 48 months, whilst the stroke score levels ranged from 15–20 (moderate stroke) and 21–42 (severe stroke), and the healthcare setting included inpatients and outpatients in palliative care at home. The CGT emerged as a theme, along with the core categories and selective coding from the transcripts. The list of respondents' backgrounds for the online in-depth interviews is presented in Table 2. All analyses were themed under core categories, as can be seen in Table 3.

**Table 2.** Respondents' background.

| No. | PTG | PTA | CGG | CGA | CD (Month) | Relationship | Religious Attendance | Religious Belief | Welfare | Stroke Levels | Healthcare Setting |
|---|---|---|---|---|---|---|---|---|---|---|---|
| EL1 | Male | 63 | Female | 61 | 24 | Spouse | Frequently | Buddhist | UCS | 21–42 | Inpatient |
| EL2 | Male | 75 | Female | 44 | 30 | Son/daughter | Daily | Buddhist | UCS | 21–42 | Outpatient |
| EL3 | Male | 77 | Male | 54 | 18 | Son/daughter | Daily | Buddhist | CSMBS | 15–20 | Inpatient |
| EL4 | Female | 87 | Male | 38 | 20 | Son/daughter | Weekly | Buddhist | UCS | 21–42 | Inpatient |
| EL5 | Female | 93 | Female | 44 | 15 | Son/daughter | Daily | Buddhist | CSMBS | 21–42 | Inpatient |
| EL6 | Male | 61 | Female | 49 | 36 | Son/daughter | Daily | Buddhist | UCS | 21–42 | Inpatient |
| EL7 | Male | 73 | Female | 53 | 40 | Spouse | Frequently | Buddhist | SP | 21–42 | Inpatient |
| EL8 | Female | 75 | Female | 34 | 24 | Son/daughter | Daily | Buddhist | SP | 15–20 | Outpatient |
| EL9 | Male | 74 | Male | 47 | 16 | Son/daughter | Frequently | Buddhist | CSMBS | 15–20 | Outpatient |
| EL10 | Male | 62 | Female | 70 | 30 | Son/daughter | Daily | Buddhist | CSMBS | 21–42 | Inpatient |
| EL11 | Male | 68 | Female | 35 | 26 | Son/daughter | Weekly | Buddhist | CSMBS | 21–42 | Inpatient |
| EL12 | Female | 65 | Female | 37 | 18 | Son/daughter | Daily | Buddhist | CSMBS | 15–20 | Outpatient |
| EL13 | Female | 63 | Male | 47 | 19 | Spouse | Daily | Buddhist | UCS | 21–42 | Inpatient |
| EL14 | Male | 66 | Female | 61 | 22 | Spouse | Daily | Buddhist | SP | 15–20 | Outpatient |
| EL15 | Male | 63 | Female | 59 | 27 | Spouse | Weekly | Buddhist | CSMBS | 21–42 | Inpatient |
| EL16 | Female | 64 | Female | 44 | 40 | Brother/sister | Frequently | Buddhist | CSMBS | 21–42 | Inpatient |
| EL17 | Female | 62 | Male | 52 | 48 | Spouse | Frequently | Buddhist | SP | 21–42 | Inpatient |
| EL18 | Male | 55 | Female | 49 | 27 | Spouse | Daily | Buddhist | SP | 15–20 | Outpatient |
| EL19 | Male | 60 | Male | 58 | 16 | Spouse | Daily | Buddhist | CSMBS | 21–42 | Inpatient |
| EL20 | Male | 63 | Female | 55 | 28 | Spouse | Daily | Buddhist | SP | 21–42 | Inpatient |
| EL21 | Male | 62 | Female | 59 | 30 | Spouse | Weekly | Buddhist | CSMBS | 21–42 | Inpatient |
| EL22 | Female | 67 | Male | 52 | 33 | Brother/sister | Weekly | Buddhist | CSMBS | 21–42 | Inpatient |
| EL23 | Male | 60 | Female | 44 | 42 | Son/daughter | Frequently | Buddhist | SP | 21–42 | Inpatient |
| EL24 | Male | 69 | Female | 39 | 37 | Son/daughter | Daily | Buddhist | UCS | 21–42 | Inpatient |
| EL25 | Female | 68 | Female | 40 | 36 | Son/daughter | Frequently | Buddhist | CSMBS | 15–20 | Outpatient |
| EL26 | Female | 60 | Female | 45 | 27 | Son/daughter | Daily | Buddhist | CSMBS | 21–42 | Inpatient |
| EL27 | Male | 63 | Male | 44 | 19 | Son/daughter | Weekly | Buddhist | SP | 21–42 | Inpatient |
| EL28 | Female | 69 | Male | 62 | 36 | Spouse | Daily | Buddhist | UCS | 21–42 | Inpatient |
| EL29 | Female | 62 | Female | 44 | 31 | Son/daughter | Frequently | Buddhist | CSMBS | 21–42 | Inpatient |
| EL30 | Female | 66 | Male | 55 | 25 | Spouse | Daily | Buddhist | CSMBS | 21–42 | Inpatient |

Note: EL = end-of-life stroke; PTG = patient gender; PTA = patient age; CGG = caregiver gender; CGA = caregiver age; SP = self-payment; UCS = universal coverage scheme; CSMBS = civil servant medical benefit scheme.

**Table 3.** The core theme and category.

| Theme | Category | Interview Confirmed Theme | | | | Interview Length |
|---|---|---|---|---|---|---|
| | | Round 1 | Round 2 | Round 3 | Round 4 | |
| *Theme 1* | *Important aspects of religious care* | | | | | 60 min |
| A1 | Religious coping | ✓ | ✓ | ✓ | ✓ | |
| A2 | Religious advice | ✓ | ✓ | ✓ | ✓ | |
| A3 | Religious suffering | ✓ | ✓ | ✓ | ✓ | |
| *Theme 2* | *Religious beliefs* | | | | | 90 min |
| B1 | Eternal life and faith | ✕ | ✓ | ✓ | ✓ | |
| B3 | Religious life | ✓ | ✕ | ✓ | ✓ | |
| B4 | Nature of reality | ✓ | ✓ | ✓ | ✓ | |
| *Theme 3* | *Personal spiritual life* | | | | | 90 min |
| C1 | Mental living | ✓ | ✓ | ✓ | ✓ | |
| C2 | Moment-to-moment | ✕ | ✓ | ✓ | ✓ | |
| C2 | Finding meaning | ✓ | ✓ | ✓ | ✓ | |
| C4 | Feeling connectedness | ✕ | ✓ | ✓ | ✓ | |
| *Theme 4* | *Religious mentality* | | | | | 100 min |
| D1 | Sensations | ✓ | ✕ | ✓ | ✓ | |
| D2 | Recognition | ✓ | ✓ | ✓ | ✕ | |
| D3 | Desires | ✕ | ✓ | ✓ | ✓ | |
| D4 | Consciousness | ✓ | ✓ | ✓ | ✓ | |
| *Theme 5* | *Religious medical care* | | | | | 120 min |
| E1 | Pain management | ✓ | ✓ | ✓ | ✓ | |
| E2 | Physiological stimulus | ✓ | ✓ | ✕ | ✓ | |
| E3 | Religious healing | ✓ | ✓ | ✓ | ✓ | |

The coding of the transcripts led to the development of open coding (see Table A1), axial coding (see Table A2), and selective coding (see Table A3). We developed the data using codes, core categories, initial themes, redefined concepts, and emerged theories, as can be seen in Table A4. The final theme was emerged theory of Buddhist religious care incorporated for end-of-life stroke patient receiving palliative care at home during the COVID-19 pandemic (see Table 4).

**Table 4.** The final themes' emerged theories.

| Emerged Theories | Respondents' Confirmations | | | | Frequency (%) |
|---|---|---|---|---|---|
| | **Round 1** | **Round 2** | **Round 3** | **Round 4** | |
| *Buddhist therapies* | | | | | |
| Coping mechanisms | ✓ | ✓ | ✓ | ✓ | 28 (93.33) |
| Religious consultation | ✓ | ✓ | ✓ | ✓ | 29 (96.66) |
| Religious counselling | ✓ | ✓ | ✓ | ✓ | 30 (100) |
| *Religious beliefs* | | | | | |
| Religious enlightenment | ✗ | ✓ | ✓ | ✓ | 26 (86.66) |
| Buddhist prescription | ✓ | ✓ | ✓ | ✓ | 30 (100) |
| Reality in life | ✓ | ✓ | ✓ | ✓ | 30 (100) |
| *Religious life satisfaction* | | | | | |
| Mental counselling | ✗ | ✓ | ✓ | ✓ | 27 (90) |
| Moments of care | ✓ | ✓ | ✓ | ✓ | 30 (100) |
| Life purpose | ✓ | ✓ | ✓ | ✓ | 30 (100) |
| Sense of connectedness | ✓ | ✓ | ✓ | ✓ | 29 (96.66) |
| *Religious mental care* | | | | | |
| Mental practice | ✗ | ✓ | ✓ | ✓ | 30 (100) |
| Mental healthcare | ✓ | ✓ | ✓ | ✓ | 30 (100) |
| Mental principles | ✓ | ✓ | ✓ | ✓ | 30 (100) |
| *Religious needs* | | | | | |
| Pain management | ✓ | ✓ | ✓ | ✓ | 27 (90) |
| Religious interventions | ✓ | ✓ | ✓ | ✓ | 29 (96.66) |
| Religious healing | ✓ | ✓ | ✓ | ✓ | 30 (100) |

*Coping mechanisms.* The majority of respondents experienced caring for end-of-life stroke patients in palliative care settings at home. The following statements demonstrate that family caregivers turned to their religion to cope with stroke care during the COVID-19 period:

> "…During COVID-19 palliative care…I prayed and prayed…I hollered…I did everything that I could do…Buddhist teaching [karma] is only one way to make me feel alive…I am so frightened…" (EL30)

> "…Sometimes there would be tears during COVID-19 impacts on stroke palliative care at home…I knew birth, old age, sickness, and death…I strictly followed Buddhist teaching [not to do any evil, to cultivate good, to purify one's heart], which helps to cope with stroke pain…" (EL5)

*Religious consultation.* Respondents stated that, during COVID-19, Buddhist practices—such as praying, following advice, and existence in practice—helped stroke patients to cope with the emotional pain. Buddhist consultation is delivered by community healthcare centres, and monks serve as counsellors, healers, and herbalists as well as deliverers of end-of-life care at home. Buddhist monks are directly involved in the promotion of religious care services between temples and at home. One section of the Buddhist cannon describes religious care service at home as being of great value for end-of-life care. Some respondents received advice on Buddhist consultations:

> "…I was advised by a monk…He touched me, as follows the four noble truths [the truth of suffering, the truth of the cause of suffering, the truth of the end

of suffering, and the truth of the path that leads to the end of suffering] to help relieve stroke suffering..." (EL27)

"...I invited a monk to deliver a sermon at home...You know, my mom felt brighter, her breathing became smoother, and she had a stable pulse..." (EL16)

*Religious counselling.* Respondents repeatedly mentioned that Buddhist advice helped them to provide mental care for end-life-life stroke patients receiving palliative care at home. The respondents stated that, to relieve suffering, they were taught to "let go" of what had happened in the past, live in the present, and move forwards in the future. In terms of understanding the physical and mental suffering of bedridden stroke patients, two respondents stated that:

"...I saw my mom cry softly...This is because she was suffering and had lived as a bedridden stroke patient for more than five years...I think she felt hurt, but no one could help her..." (EL8)

"...As a bedridden stroke patient [my mom] finds it so difficult to breathe...I feel she is silent and suffering from acute ischemic stroke...I only pray for her quick recovery..." (EL25)

The respondents reported that stroke suffering is associated with feelings of hurt, fear, anxiety, and failure. They explained that "suffering is unseen", and is not only about confronting the possibility of being a stroke patient, but also about keeping open the potential for future responsibilities. Family caregivers identified realistic hopelessness in caring, the emotional suffering, and the struggle inherent in recovery from acute ischemic stroke faced by patients at home.

*Religious beliefs.* The respondents referred to religious beliefs related to eternal life and faith, the purpose of life, and the nature of reality. They discussed religious beliefs in the Buddha teaching, as seen via the following statements:

"...Eternal life and faith in Buddhism is a cornerstone of relief for stroke patient practices in palliative care during the COVID-19 pandemic...Yet, Buddha teaching has been designed to help us let go completely—meditation, chanting, prayer, ritual, and sutras..." (EL11)

"...The life purpose is to achieve enlightenment...You know, understanding that refuge in the Buddha, refuge in the Dharma, and refuge in the Sangha is the key point for relieving the root of suffering from stroke..." (EL29)

"...The nature of reality [the stroke condition is severe [soul] pain and disability...The only way I accepted it is that it is the truth of the cause of suffering...I think beliefs, follows, and practices in Buddhist teaching help us..." (EL13)

As seen in the above quotes, the respondents stated that eternal life and faith in Buddhism provided strength, hope, belief, and aliveness. The key elements of Buddhist teaching are the truth of the cause of suffering and the truth of the end of suffering to relieve stroke pain, which is essential for Buddhist patients.

*Religious life satisfaction.* The majority of respondents viewed personal religious life as being associated with mental living for patient counselling, living in the moment, finding meaning, and connectedness. Four respondents expressed that mental living is one of the important aspects of religious life satisfaction when receiving palliative care at home during the COVID-19 pandemic:

"...I cared for my mom who had COVID-19 complications and it was so stressful...I felt as if her mind was being suffocated when the nurse gave her oxygen...I am not sure how long it will take for her to recover...I only tell her to pray every day..." (EL26)

"...It was draining when I saw my dad in the bed during corona times...I was so stressed while my dad's brain tissue was being starved of life-giving oxygen...At

that moment, I had a million questions; until now, there has been no one to answer me..." (EL23)

"...I was grateful to talk and find meaning in Buddhist practices [doing, being, and the rest] which helped relieve my mom's suffering due to her stroke..." (EL5)

"...I feel I connected in Buddhist life as in birth...I detach from the wholeness of the true inner self...You know, the key point of the Buddha's teaching is 'beyond oneself', reflecting on how I have lived with stroke patients at home..." (EL19)

The respondents expressed that, for the family caregiver, and for the stroke patients themselves, there was mental suffering, healing seriousness, and emotional stress during COVID-19 complications. They suggested that Buddhist principles (meditation, wisdom, compassion, and wholeness) helped to relieve the suffering from stroke pain.

*Religious mental care.* Religious mentality refers to sensations in Buddhism (vedanā), which means that internal sense organs come into contact with external sense objects and the associated consciousness. Recognition (saññā) in Buddhism refers to mental formation. And desires (taṇhā) in Buddhism refer to craving pleasure, material goods, and immortality. The majority of respondents identified the core values of Buddhist religious mentalities (commonly known as Buddhist mental care) as being related to concentration, medication, and mental development. They noted that religious mentalities commonly revolve around sensations, recognition, desires, and consciousness, which blur the stroke pain during the COVID-19 pandemic. Four respondents, in particular, suggested that:

"...With Buddhism, sensations of internal and external pain from the causes of stroke suffering...One thing I can do...I pray for her...I hope she will recover soon..." (EL21)

"...I linked lives with COVID-19 stroke patients for a year...I have never seen my dad talk, smile, and cry with me...In the last month, he had injected medicine...I felt he was not recognised for medical treatment..." (EL3)

"...I carried my wife at home...She told me she needed to die in her place of birth...This was her desire...She accepted her physical pain..." (EL17)

"...Most of the home stroke care is strong consciousness [mind]...For instance, my husband has a very strong mind [mental power] and body [soul] which helped him cope with fear..." (EL1)

Interestingly, they understood the internal and external pain, as well as the medical recognition related to the alternative stroke treatment of religious care. The respondents also stated that the core concepts of Buddhist teaching, such as consciousness, past tendencies, and delusion, provided relief from suffering.

*Religious needs.* The respondents asserted that religious medical care is built on pain management, physiological stimulus, and religious healing. They stated that Buddhist teaching helps patients to cope with stroke pain, emotional stress, and pain control choices. Three respondents succinctly explained how to cope with stroke pain when receiving palliative care at home during COVID-19 complications:

"...Buddhists believe that stroke pain is karmic...In particular, if we suffer in the current life, it is due to negative action...Likewise, if we prosper, it is due to past positive acts, such as compassion...Treating karmic disease is to provide relief [my husband] pain calmly, without becoming emotionally distressed..." (EL15)

"...Physiological stimulus for stroke patients has played an important role in medical treatment...I closely intertwined medicine with Buddhist ritual care to relieve his stroke pain during the coronavirus pandemic..." (EL27)

"...I tried to alleviate [my husband's] stroke pain...The monk trained me how to cope with religious healing [ritual and prayer] for relieving physical suffering while caring at home..." (EL1)

The respondents indicated that religious medical care catered primarily to Buddhist principles for recovering from physiological stimulus and achieving pain management as well as religious healing. Family caregivers indicated that, whilst conducting palliative care at home, Buddhist teachings helped them to deal with the medical conditions and provided freedom from anxiety during the COVID-19 complications.

Most respondents concluded that Buddhist religious care is incorporated for end-of-life stroke patients receiving palliative care at home during the COVID-19 pandemic in central Thailand. Our CGT study illustrated that coping mechanisms, religious consultation, and religious counselling are involved in Buddhist therapies. We found that religious beliefs are associated with religious enlightenment, Buddhist prescription, and reality in life. Ongoing longitudinal follow-up areas emerged, such as mental counselling, moments of care, life purpose, and sense of connectedness encouraged to achieve religious life satisfaction. The respondents summarised that religious mental care (mental practice, healthcare, and principles) and religious needs (pain management, religious intervention, and healing) have been incorporated for end-of-life stroke patients receiving palliative care at home during COVID-19 complications.

## 4. Discussion

### 4.1. Discussion of Findings

Our respondents endeavoured to answer *Research Question 1*, which enabled us to discuss the important aspects of religious care during the COVID-19 pandemic. We utilised the CGT approach to assess Buddhist religious care incorporated for end-of-life stroke patients receiving palliative care at home. Via a longitudinal in-depth interview from June 2020 to March 2022, our CGT study showed that Buddhist therapies incorporated during palliative care were associated with coping mechanisms, religious consultation, and religious counselling. This finding is consistent with earlier studies on religious coping with the COVID-19 pandemic (Counted et al. 2022; Wilt et al. 2022; Xu 2021), wherein meaning-making coping, meditative coping, and ego-transcendence coping were unveiled. Clearly, respondents addressed Buddhist therapies: that is, they often barely perceived suffering because of the onset and magnitude of stroke pain and the pandemic crisis (Bentzen 2021; Best et al. 2016; Diego-Cordero et al. 2022; Kwak et al. 2022; Zhang et al. 2021).

Respondents in the three sites are prone to stroke suffering due to the first and second wave of the COVID-19 pandemic. Despite the encouragement of preventative COVID-19 measures, access to essential religious consultation, counselling, and Buddhist-orientated coping is still unravelling. This finding may help to explain why end-of-life stroke patients and family caregivers engage with Buddhist religious care during the COVID-19 pandemic (Upasen et al. 2022; Wiseso et al. 2022; Wisesrith et al. 2021). The results underscore how the entwining of religious coping and suffering results in the kinds of palliative care delivered during COVID-19 (Connolly and Timmins 2022; Domaradzki 2022; Gilissen et al. 2020). Interestingly, we also observed how the value attributed to Buddhist therapies seems to vary depending on the coping mechanisms (praying, control, and mental living), which reduces ischemic stroke suffering during the COVID-19 pandemic. There is thus potential for Buddhist therapies given their effectiveness in reducing depressive symptoms, craving for existence, and things in the presence of oneself.

Regarding the respondents' answers to *Research Question 2*, our CGT study obtained a longitudinal in-depth interview in four time periods, and the respondents identified four themes: religious beliefs, religious life satisfaction, religious mental care, and religious needs during the COVID-19 pandemic. This result is consistent with the studies of Bentzen (2021), Counted et al. (2022), Domaradzki (2022), Soonthornchaiya (2020), and Upasen et al. (2022), which showed that family caregivers involve religious life satisfaction in their ability to relieve stroke pain during the COVID-19 pandemic. As described in the findings, end-of-life family caregivers who experienced mental counselling, life purpose, and moments of care during the early COVID-19 pandemic period were very stressed about providing stroke care at home.

Our respondents explained that Buddhist religious care is essential for consciousness, sense of connectedness, sensations, and desires. Besides the findings regarding the proximate relationship of the self, family caregivers, and close ones (Chaiwutikornwanich 2015; Diego-Cordero et al. 2022; Jung et al. 2022; Mamom et al. 2020; Niu et al. 2020; Strong 2021), it is important for stroke patients to receive palliative care at home. Our findings indicated that crucial mental practice (meditation), healthcare (counselling), and principles (calling into existence) reinforce the relieving of stroke pain during the COVID-19 pandemic. Buddhists' existences have reduced stroke pain via religious healing, addiction recovery, palliative sedation, and the end of suffering. This finding is consistent with Maddock (2022), who saw understanding as suffering, non-self (unchanging), and impermanence; indeed, understanding is one of the core Buddhist existences. Buddhist religious care may be more easily accommodated than palliative care, as it is provided as per a cognitive structure (need for closure) and involves ritualised practices and restrictions (a basic creed of Buddhism).

It is a subject of ongoing debate how—or even if—Buddhist religious practices help to cope with stroke pain, spontaneous awakenings, and the task of religious healing during the COVID-19 pandemic (Connolly and Timmins 2022; Dorji and Lapierre 2022; Gilissen et al. 2020; Kwak et al. 2022; Maddock 2022; Thambhitaks and Boonyathee 2021; Wilt et al. 2022). However, our current study, and previously published work in Thailand (Marshall et al. 2021; Nilmanat 2016; Sethabouppha and Kane 2005; Soonthornchaiya 2020; Sukcharoen et al. 2020; Taniyama and Becker 2014; Upasen et al. 2022), indicate that Buddhist religious needs are typically used to cope with pain management, intervention, and religious healing incorporated for end-of-life stroke care at home during COVID-19 complications. A direct expression of family caregivers provides Buddhist religious care with clear ways of coping with suffering, mental wellness, and ritual care, which makes it possible to achieve a deep insight and consciousness.

*4.2. Theoretical Contributions*

This article makes several theoretical contributions. Our first theoretical contribution is to the existing Buddhist therapy incorporated for end-of-life stroke patients receiving palliative care (Gray 2020). Our respondents showed that Buddhist therapies play an important role in coping mechanisms, religious consultation, and religious counselling. Thus, we contribute to the scarce literature on Buddhist therapy for stroke patients (Stewart et al. 2016), Buddhist religious coping (Falb and Pargament 2013), and Thai Buddhist family caregivers (Wiseso et al. 2022). In doing so, the present study also adds to the existing body of research on the Buddhist therapies coping framework for end-of-life stroke patients at home (Pilaikiat et al. 2016) by relying on grounded theoretical data.

Our second theoretical contribution to Buddhist religious beliefs, as demonstrated in the literature (O'Brien et al. 2018; Wisesrith et al. 2021), showed that patients' beliefs, faith, and religious rituals constitute one component of efficient holistic palliative care. We contributed to filling the gaps in grounded data on Buddhist religious care for enlightenment, Buddhist prescription, and life purpose (Okamura et al. 2018). Praying consistently improved Buddhist beliefs about eternal life and faith, religious life, and the nature of stroke diseases. Previous studies indicated that Buddhists' recognition of personal autonomy, freedom of choice, and mental aid relieve stroke pain (Dorji and Lapierre 2022). This explains how palliative caregivers apply Buddhist religious beliefs as part of coping, advice, and suffering (Chow et al. 2021).

Our third theoretical contribution is to the religious life satisfaction of end-of-life stroke patients receiving palliative care at home. We found that mental living, interpersonal interactions, a sense of purpose, and living in the moment care during COVID-19 complications were crucial. These findings are consistent with Chaiwutikornwanich (2015), Falb and Pargament (2013), Pilaikiat et al. (2016), and Stewart et al. (2016), who found that Buddhist religious care is related to religious life satisfaction in palliative care. For example, Maddock (2022) demonstrated that the Buddhist model can be used to mitigate

the cognitive, emotional, and physical demands of family caregivers. Our study found that Buddhist religious life satisfaction was experienced by end-of-life stroke patients receiving palliative care at home, as well as by their caregivers.

Our fourth theoretical contribution is to support the findings of Stewart et al. (2016) and Taniyama and Becker (2014), regarding the importance of religious practices, reciting prayers, making merit, and meditation. Our respondents identified that the religious mental care of end-of-life stroke patients is associated with sensations, recognition, desires, and consciousness. Previous studies (Shonin et al. 2014) indicated that delusion, non-self, nonattachment, interconnectedness, and emptiness are parts of Buddhist mental care. According to Dorji and Lapierre (2022), nature, peace, and consciousness are important to religious mental care during palliative care at home. We assert that refuge in the mental practice, healthcare, and principles is associated with stroke recovery during the COVID-19 pandemic.

Our final theoretical contribution is to the religious needs arising from pain management, physiological stimulus, and religious healing. We add to the findings of previous studies (Maddock 2022) regarding the Buddhist self-care of end-of-life stroke patients receiving palliative care at home. Buddhist religious needs are attributed to spontaneous awakenings and coping with pain and suffering during the COVID-19 pandemic (Dhavernas and Williams 2022). From grounded data, Niu et al. (2020) found that Buddhist religious needs emphasised the need for patient-centred care to cope with pain, religious interventions, and healing. Essentially, we found that Buddhist religious needs are relieved in pain management, physiological stimulus, and medical conditions.

### 4.3. Practical Implications

This article has several practical implications. First and foremost, as our CGT study suggests, Buddhist religious care has a large immaterial value for end-of-life stroke patients receiving palliative care at home during the COVD-19 pandemic. In terms of this value, it is a potential Buddhist therapy for coping mechanisms, religious consultation, and religious counselling which will help to relieve stroke suffering. Second, religious enlightenment, prescription, and reality in life are substantiated to cope with mental and physical pain. Third, religious life satisfaction helps patients and caregivers through the increase in mental counselling, moments of care, and life purpose. Fourth, religious mental care can find ways to mental practice, healthcare, and principles which incorporate Buddhist care during the COVID-19 period. Finally, end-of-life strokes patients foster religious needs, which enable them to cope with pain management, religious interventions and religious healing at the time of the COVID-19 pandemic.

### 4.4. Limitations and Future Studies

This article has some limitations. Firstly, in keeping with theoretical sampling, a total of 30 respondents were recruited before saturation. This may appear to be a small sample in terms of patient representation in central Thailand. Secondly, the CGT method was conducted at three sites—namely the Angthong, Ayutthaya, and Pratumthani provinces—meaning that other regions may not be represented. Thirdly, online interviews were re-conducted in four rounds with the same respondents, thereby challenging the data collection during the COVID-19 pandemic. Fourthly, although the respondents provided some valuable insights, their experiences cannot be generalised, as they may not reflect the wide Buddhist religious care and palliative care realities. Finally, future studies should conduct quantitative (framework, hypothesis, and sample size) research to test the effect of Buddhist religious care during the post-COVID-19 period, so that findings can be generalised using the empirical evidence.

**Author Contributions:** Conceptualisation, J.M. and H.D.; methodology, J.M. and H.D.; formal analysis, J.M. and H.D.; writing—original draft preparation, J.M. and H.D.; writing—review and editing, J.M. and H.D.; project administration, J.M. and H.D.; funding acquisition, J.M. and H.D. All authors have read and agreed to the published version of the manuscript.

**Funding:** This research was funded by the National Research Council of Thailand [Grant No. N72B640120].

**Institutional Review Board Statement:** The study was conducted following the Institutional Review Board at the Thammasat University and the Declaration of Helsinki, the Belmont Report, CIOMS Guidelines and International Practice (ICH-GCP) (COA No. 117/2562, Ref No. 065/2562). All respondents were fully informed of the study objectives, content, procedures, and principles of confidentiality.

**Informed Consent Statement:** Written informed consent has been obtained from the patient(s) to publish this paper.

**Data Availability Statement:** Not applicable.

**Conflicts of Interest:** The authors declare no conflict of interest.

## Appendix A

The first example involved conducting open coding to extract, describe, and classify the interview transcripts (see Table A1). The second example involved the development of axial coding to establish the logical links between conceptual descriptions and their relations (see Table A2). The third example included selective coding as iterative to a core category, systematically relating to a sub-category, filling the category, and emerging theories (see Table A3). The final example placed conceptual labels on the initial themes, redefined concepts, and constructed theory (see Table A4).

**Table A1.** An example of open coding.

| Open Coding | Examples of Respondents' Quotations |
|---|---|
| End-of-life stroke receiving palliative care | "…I defined end-of-life stroke as constituting a long-term coma, symptoms, terminally ill, and physical disability receiving palliative care at home…" (EL7) |
| Religious care | "…I think religion care is an alternative choice to cope with physical and mental suffering from stroke patients..." (EL10) |
| | "…I am Buddhist…I follow and practice the teaching principles, which can help me to control myself throughout my life …" (M18) |
| Religious belief | "…I think that religious care helps with existential thoughts and living with stroke pain…" (EL1) |
| Personal religious life | "…Religious life follows Buddhist teachings, such as meditation, religious labour, and good behaviour…" (EL15) |
| Religious mentality | "…I adhere strictly to chanting, praying, and following Buddha teaching…I recite Buddhist scriptures for worship, meditation, and awareness…." (EL20) |
| Religious medical care | "…Buddhist health cares are the key features of right effort, right mindfulness and right concentration to treat me during medical care…" (EL28) |

**Table A2.** An example of axial coding.

| Axial Coding | Examples of Respondents' Quotations |
|---|---|
| The truth of suffering | "…I cared for my mom more than one year ago…I think if she can talk, drink, and eat…She can live with us…This is the way it is when someone has had a stroke…" (EL12) |
| Religious practice | "…I prayed together with my mom…Basically, we found refuge in the Buddha, refuge in the Dharma, and refuge in the Sangha to relieve the stroke pain…" (EL4) |
| Practical application | "…I learned how to cope with stroke…This is part of life…Buddhist healthcare is to alleviate, prevent, and relieve suffering…" (EL22) |
| Religious medical care | "…Stroke is likely karmic life…Improving self-confidence is the core mindfulness-based stress reduction for medical care…" (EL14) |

**Table A3.** An example of selective coding.

| Selective Coding | Examples of Respondents' Quotations |
|---|---|
| Eternal life and faith | "...I am a Buddhist person...It is a holy spirit of my eternal life and faith in teaching and practices...There is no permanent self, but I have to live in the present..." (EL9) |
| Mental living | "...I care that my mom [giving birth] brought to me to happiness...I am not only a stroke caregiver, but it is my tribute..." (EL24) |
| Sensations | "...Mindfulness-based cognitive therapy for stroke patients is understood through their thoughts, emotions, awareness, and sensations to relieve the pain..." (EL6) |
| Coping with pain | "...Buddhist practices help to cope with suffering ...Chanting and meditation may help to control stroke pain ..." (EL2) |

**Table A4.** Example of initial themes, redefined concepts, and constructive theories.

| Sub-Category Themes | Redefined Concepts | Constructive Theories |
|---|---|---|
| *Important aspects of religious care* | *Religious healthcare* | *Buddhist therapies* |
| Religious coping | Coping mechanisms | Coping mechanisms |
| Religious advice | Religious consultation | Religious consultation |
| Religious suffering | Religious suffering | Religious counselling |
| *Religious beliefs* | *Religious beliefs* | *Religious beliefs* |
| Eternal life and faith | Life after death | Religious enlightenment |
| Purpose of life | Life desires | Buddhist prescription |
| Nature of reality | Reality in life | Reality in life |
| *Religious life* | *Meaningful life* | *Religious life satisfaction* |
| Mental living | Mental wellness | Mental counselling |
| Moment-to-moment | Social interactions | Moments of care |
| Finding meaning | Life purpose | Life purpose |
| Feeling connectedness | Sense of Buddhist belonging | Sense of connectedness |
| *Religious mentality* | *Religious practices* | *Religious mental care* |
| Sensations | Buddhist practice | Mental practice |
| Recognition | Buddhist practices | Mental practices |
| Desires | Buddhist healthcare | Mental healthcare |
| Consciousness | Buddhist teaching | Mental principles |
| *Religious medical care* | *Religious attention* | *Religious needs* |
| Coping with pain | Living hope | Pain management |
| Physiological stimulus | Spontaneous awakenings | Religious interventions |
| Religious healing | Religious healing | Religious healing |

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
