# Peer review of "How Buddhist Religious Care Is Incorporated for End-of-Life Stroke Patients Receiving Palliative Care at Home during the COVID-19 Pandemic: Revisiting Constructivist Grounded Theory"

_religions, doi:10.3390/rel13101000_

Round 1

Reviewer 1 Report

This was an interesting and well-written paper about an important topic. The insights seem practical and useful.

It's important to note that I have not done extensive qualitative research, so I would defer to other reviewers in terms of the details of research design.

My only suggestions center around the Introduction, which was extremely brief. I think that more could be done here to set the stage for the reader and give them more grounding in the relevant concepts.

----introduce just a few key concepts and references about Buddhism

----mention the literature on religious coping (e.g., Ken Pargament's work) and briefly cite any work that you can find related to: a) religious coping in a Buddhist framework (or with primarily Buddhist samples), b) religious coping with COVID-19, and c) religious coping in relation to caregiving. 

Author Response

Response to reviewers’ comments

Dear Reviewer,

Thank you for reviewing the manuscript entitled “How Buddhist Religious Care is Incorporated for End-of-Life Stroke Patients Receiving Palliative Care at Home during the COVID-19 Pandemic: Revisiting Constructivist Grounded Theory”. We have now addressed your comments and the new changes in the manuscript are highlighted in yellow in the text. We believe the manuscript is now stronger and clearer.

Sincerely,

The author

==========================================

REVIEWER COMMENTS FOR THE AUTHOR:

Response to reviewer #1 

Point 1:

This was an interesting and well-written paper about an important topic. The insights seem practical and useful.

Response 1:

Thank you for constructive comments, which have helped us to strengthen our paper.

Point 2:

It's important to note that I have not done extensive qualitative research, so I would defer to other reviewers in terms of the details of research design.

Response 2:

Thank you for taking the time to carefully review our manuscript. We have extensively modified the manuscript in order to fulfil your suggestions.

Point 3:

My only suggestions center around the Introduction, which was extremely brief. I think that more could be done here to set the stage for the reader and give them more grounding in the relevant concepts.

Response 3:

Thank you very much for this useful comment. In our revised version, we have added and modified the introduction section as suggested. See page 1-2 (line 21-100).

Point 4:

----introduce just a few key concepts and references about Buddhism.

Response 4:

Thank you for the comment and for critically analyzing our introduction. We have now expanded and modified the introduction as suggested. First, we have added the substantial key concepts and references related to Buddhism. Second, we have also reinforced the importance of investigating the Buddhist religious care (see line 21-100, page 1-2).

Point 5:

----mention the literature on religious coping (e.g., Ken Pargament's work) and briefly cite any work that you can find related to: a) religious coping in a Buddhist framework (or with primarily Buddhist samples), b) religious coping with COVID-19, and c) religious coping in relation to caregiving.

Response 5:

Thank you for your detailed revision of our manuscript. In our revised version, we have now added the further information on religious coping in Buddhist framework (page 2, line 72-80), religious coping with COVID-19 (see page 2, line 62-71), and religious coping in relation to caregiving (page 2, line 72-80).

Reviewer 2 Report

Add that the study was reviewed and approved by review board.

Author Response

Response to reviewers’ comments

Dear Reviewer,

Thank you for reviewing the manuscript entitled “How Buddhist Religious Care is Incorporated for End-of-Life Stroke Patients Receiving Palliative Care at Home during the COVID-19 Pandemic: Revisiting Constructivist Grounded Theory”. We have now addressed your comments and the new changes in the manuscript are highlighted in yellow in the text.

Sincerely,

The author

==========================================

REVIEWER COMMENTS FOR THE AUTHOR:

Response to reviewer #2

Point 1:

Add that the study was reviewed and approved by review board.

Response 1:

Thank you for taking the time to carefully review our manuscript. We have extensively modified the manuscript in order to fulfil your suggestion. In our revised version, we have now added the ethical consideration into the method section (sub-section, 2.3. Ethical Considerations) (see line 128-136, page 4). See our sample of ethical approval as “The Human Research Ethics Committee of Thammasat University (Science), with the first author is affiliated ethical approval for this study. This article was approved by the Thammasat University and the Declaration of Helsinki, the Belmont Report, CIOMS Guidelines and International Practice (ICH-GCP) (COA No. 117/2562, Ref No. 065/2562). All respondents were informed about the research objective, participated voluntarily, and signed a formed consent document. The respondent’s anonymity, privacy, confidentiality, and personalized information were guaranteed, which obtained or used at any stage of the study. All data were de-identified and reported only at an aggregate level.”

Reviewer 3 Report

This is an interesting method for study of Buddhism as a lived religion, how people actually appropriate and implement it. It contrasts significantly with many idealized and abstract presentations of Buddhist doctrines, with which Western psychologists try to work as Buddhist clinicians. The setting and particular situations of the subjects of this paper are as practical and down to earth as one could get. Buddhism here comes into focus as a set of mental-control strategies, or basic beliefs about suffering and its meaning. The emphasis, almost exclusively, is on dharma, and on individual practices of meditation and mindfulness. The Buddha as a personality, or role model, plays no role whatsoever- which may be perfectly accurate as far as how Buddhism is lived out by these care-takers. The sangha, however, does appear. Priests and monks make pastoral visits, although there are no other references to a religious support community. I do not know if this too accurately reflects the way Buddhism emphasizes interior mental practices rather than interpersonal relationships. I may be contrasting this with what a paper on the same topic done among Christians might yield. "What would Jesus do?" and the role of the church fellowship would be prominent, while there seem to be no Buddhist equivalents.

Author Response

Response to reviewers’ comments

Dear Reviewer,

Thank you for reviewing the manuscript entitled “How Buddhist Religious Care is Incorporated for End-of-Life Stroke Patients Receiving Palliative Care at Home during the COVID-19 Pandemic: Revisiting Constructivist Grounded Theory”. We have now addressed your comments and the new changes in the manuscript are highlighted in yellow in the text.

Sincerely,

The author

==========================================

REVIEWER COMMENTS FOR THE AUTHOR:

Response to reviewer #3

Point 1:

This is an interesting method for study of Buddhism as a lived religion, how people actually appropriate and implement it. It contrasts significantly with many idealized and abstract presentations of Buddhist doctrines, with which Western psychologists try to work as Buddhist clinicians. The setting and particular situations of the subjects of this paper are as practical and down to earth as one could get. Buddhism here comes into focus as a set of mental-control strategies, or basic beliefs about suffering and its meaning. The emphasis, almost exclusively, is on dharma, and on individual practices of meditation and mindfulness. The Buddha as a personality, or role model, plays no role whatsoever- which may be perfectly accurate as far as how Buddhism is lived out by these care-takers. The sangha, however, does appear. Priests and monks make pastoral visits, although there are no other references to a religious support community. I do not know if this too accurately reflects the way Buddhism emphasizes interior mental practices rather than interpersonal relationships. I may be contrasting this with what a paper on the same topic done among Christians might yield. "What would Jesus do?" and the role of the church fellowship would be prominent, while there seem to be no Buddhist equivalents.

Response 1:

Thank you for taking the time to carefully review our manuscript. Thank you for this comment and for pointing out that we have to clarify this further. We have extensively modified the manuscript in order to fulfil your suggestions. First, we have now extensively modified the abstract section (see our abstract section, line 6-17). For instance, we modified the term of Buddhist doctrine as Buddhist therapies. Second, we have extensively rewritten the introduction as Buddhist religious care (Kongsuwan et al. 2010; Sethabouppha and Kane 2005; Strong 2021), focused on following, teaching, and practicing. Moreover, our study was adopted from Smith-Stoner (2005) incorporating Buddhist religious care is associated with belief system (basic practices), personal religiosity (learn how the patient functions), integrated with a religious community (activities in the religiosity), ritualized practice and restrictions (practiced by Sangha members), implications for medical care (advance directives), and terminal event planning (read the scripts to the patient). And see our new version of revised version (line 21-100, page 1-3). Third, we have accordingly modified our methods (see line 101-183, page 3-5). Fourth, we have added and rewritten the discussion of findings and the participants responded to research questions in four times (see line 314-367), and finally, we have now included the further practical implications (see line 432-442).